# Brain tumours repurpose endogenous neuron to microglia signalling mechanisms to promote their own proliferation

**Kelda Chia, Marcus Keatinge, Julie Mazzolini, Dirk Sieger***

Centre for Discovery Brain Sciences, University of Edinburgh, Edinburgh, United Kingdom

**Abstract** Previously we described direct cellular interactions between microglia and AKT1+ brain tumour cells in zebrafish (Chia et al., 2018). However, it was unclear how these interactions were initiated: it was also not clear if they had an impact on the growth of tumour cells. Here, we show that neoplastic cells hijack mechanisms that are usually employed to direct microglial processes towards highly active neurons and injuries in the brain. We show that AKT1+ cells possess dynamically regulated high intracellular $Ca^{2+}$ levels. Using a combination of live imaging, genetic and pharmacological tools, we show that these $Ca^{2+}$ transients stimulate ATP-mediated interactions with microglia. Interfering with $Ca^{2+}$ levels, inhibiting ATP release and CRISPR-mediated mutation of the *p2ry12* locus abolishes these interactions. Finally, we show that reducing the number of microglial interactions significantly impairs the proliferation of neoplastic AKT1 cells. In conclusion, neoplastic cells repurpose the endogenous neuron to microglia signalling mechanism via P2ry12 activation to promote their own proliferation.
DOI: https://doi.org/10.7554/eLife.46912.001

## Introduction

Microglia and infiltrating macrophages are amongst the most abundant cell types in the microenvironment of brain tumours and have been shown to actively promote tumour growth (*Hambardzumyan et al., 2016*; *Quail and Joyce, 2017*). A variety of mechanism contribute to this tumour-promoting activity including modifications of the extracellular matrix, the induction of angiogenesis and the generation of an immunosuppressive environment (*Markovic et al., 2005*; *Komohara et al., 2008*; *Wu et al., 2010*; *Zhai et al., 2011*; *Zhang et al., 2012*; *Ellert-Miklaszewska et al., 2013*; *Pyonteck et al., 2013*; *Wang et al., 2013*; *Hambardzumyan et al., 2016*).

Intriguingly, direct cellular interactions between microglia and brain tumour cells have been described (*Bayerl et al., 2016*; *Hamilton et al., 2016*; *Resende et al., 2016*; *Ricard et al., 2016*). These cellular interactions consist of different types of direct surface contacts between microglia and tumour cells, from microglia constantly extending processes towards tumour cells to microglia flattening their surfaces around tumour cells. Importantly, these interactions were long-lasting and did not appear to be anti-tumoural as phagocytic events were not observed. These observations have been consistently made in a variety of models of orthotopic transplantations of human and mouse glioma cells into mouse or zebrafish brains (*Bayerl et al., 2016*; *Hamilton et al., 2016*; *Resende et al., 2016*; *Ricard et al., 2016*). Furthermore, we have recently shown that these cellular interactions are initiated during the earliest stages of tumour growth as they can already be observed between microglia and pre-neoplastic AKT1+ cells (*Chia et al., 2018*). However, the signals that initiate these cellular interactions and their functional impact on tumour cells have not been addressed so far.

*For correspondence:
dirk.sieger@ed.ac.uk

Competing interests: The authors declare that no competing interests exist.

Several elegant studies have described cellular interactions between microglia and neurons under physiological conditions. Microglia have been observed to direct cellular processes towards neurons with increased intracellular $Ca^{2+}$ levels (*Li et al., 2012*; *Sieger et al., 2012*; *Eyo et al., 2014*; *Eyo et al., 2015*). These interactions were regulated by ATP/ADP released from the neurons upon intracellular $Ca^{2+}$ increase and sensed by the microglia via the purinergic P2y12 receptor (*Li et al., 2012*; *Sieger et al., 2012*; *Eyo et al., 2014*; *Eyo et al., 2015*).

Here, we hypothesised that mechanisms employed by healthy neurons to attract microglial processes are hijacked by neoplastic cells to stimulate interactions and that these interactions promote the growth of neoplastic cells. To address these questions, we made use of our recently published zebrafish brain tumour model to analyze interactions between microglia and neoplastic AKT1 over-expressing cells (*Chia et al., 2018*). We show that AKT1+ cells have significantly increased $Ca^{2+}$ levels, which are dynamically regulated. Pharmacological inhibition of NMDA receptor signalling significantly decreased $Ca^{2+}$ levels in AKT1+ cells and drastically reduced the number of microglial interactions with these cells. In line with these results, inhibition of ATP release and knock out of the *p2y12* receptor abolished microglia interactions with AKT1+ cells, showing that $Ca^{2+}$-mediated ATP signalling is required for these cellular contacts. Intriguingly, we showed that reducing these interactions had a direct functional impact on AKT1 cells and reduced their proliferative capacities.

## Results

### Microglia closely interact with pre-neoplastic AKT1 cells

We and others have shown previously that microglia show direct cellular interactions with tumour cells and pre-neoplastic AKT1+ cells in the brain (*Bayerl et al., 2016*; *Hamilton et al., 2016*; *Resende et al., 2016*; *Ricard et al., 2016*; *Chia et al., 2018*). However, the underlying mechanisms promoting these interactions have not been identified. Here we analysed these cellular contacts between microglia and pre-neoplastic AKT1+ cells in more detail. To induce AKT1 expression in neural cells we followed the previously published strategy by expressing AKT1 under the neural-specific beta tubulin (NBT) promoter using a dominant active version of the LexPR transcriptional activator system (ΔLexPR) (*Chia et al., 2018*). We co-injected an NBT:ΔlexPR-lexOP-pA driver plasmid together with a lexOP:*AKT1*-lexOP:tagRFP construct into mpeg1:EGFP transgenic zebrafish in which all macrophages including microglia are labelled (*Figure 1*) (*Ellett et al., 2011*; *Chia et al., 2018*). Control fish were injected with a lexOP:tagRFP construct. In this model, cellular abnormalities and increased proliferation are detected in AKT1+ cells within the first week of development and solid tumours can be observed from 1 month of age (*Chia et al., 2018*). As described previously, microglia were observed to cluster in areas of AKT1+ cells while their distribution appeared normal in fish injected with the lexOP:tagRFP control construct (*Figure 1A,B*, *Figure 1—videos 1* and *2*). Furthermore, direct cellular interactions between microglia and AKT1+ cells seemed to be more frequent compared to interactions between microglia and RFP control cells. Thus, we decided to analyze these interactions in more detail and quantified these interactions. We counted the number of microglia in direct contact with AKT1+ or RFP control cells and normalised by the total number of microglia in the respective sample. Importantly, microglia showed a significantly increased number of direct interactions with AKT1+ cells compared to control cells (*Figure 1C*). As described before, different types of interactions were observed which ranged from microglia extending processes towards AKT1+ cells to microglia flattening and moving their cellular surface around AKT1+ cells (*Figure 1D,E*, *Figure 1—videos 3* and *4*). Furthermore, two or more microglial cells were frequently observed to interact with the same AKT1+ cell (*Figure 1E*, *Figure 1—video 4*). Interestingly, overexpression of HRASV12 in neural cells as well as overexpression of AKT1 and HRASV12 under control of the zic4 enhancer stimulated similar microglial responses (*Figure 1—figure supplement 1*). Thus, cells in the brain undergoing oncogenic transformation via AKT1 and HRASV12, seem to possess signals stimulating these long-lasting interactions.

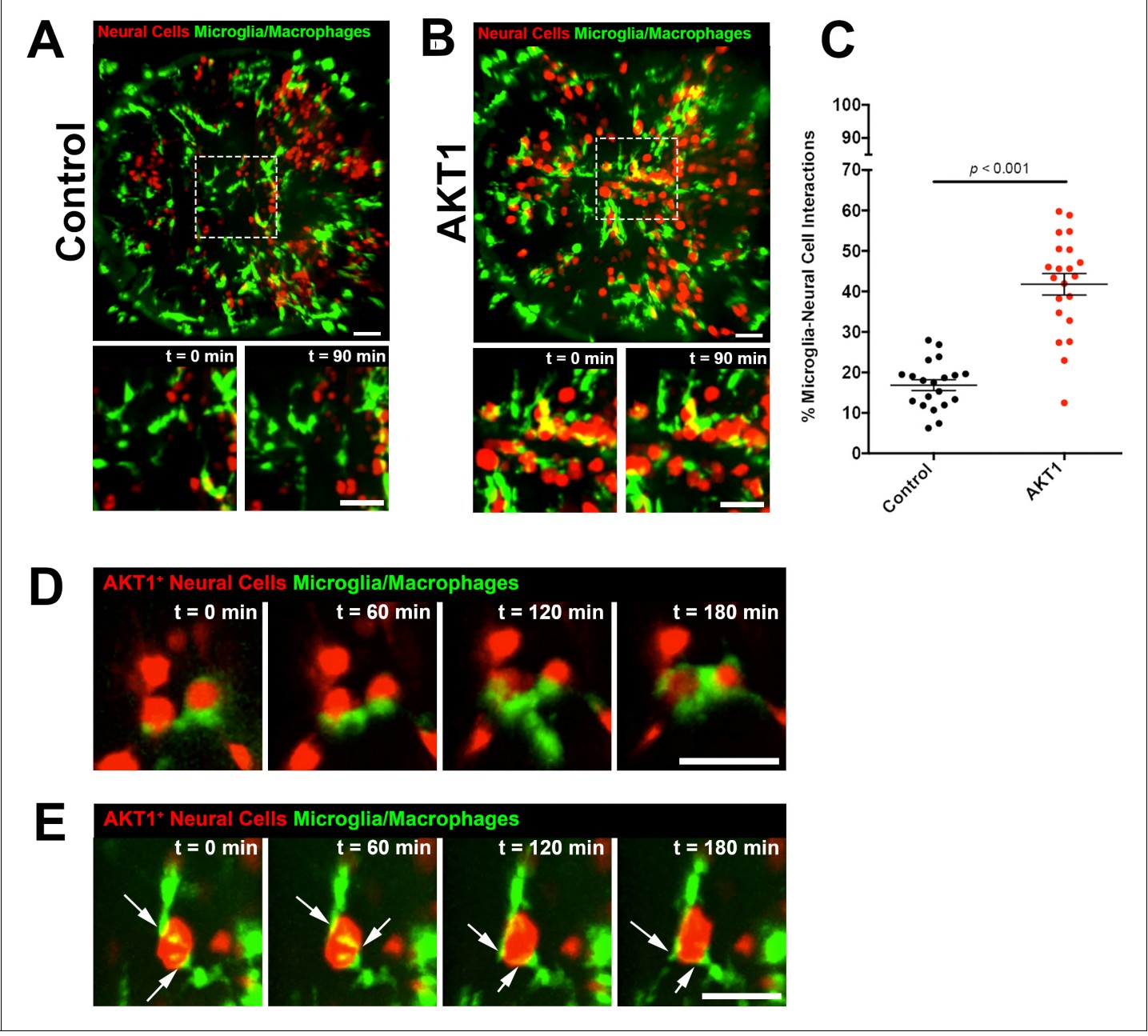

**Figure 1.** Microglia show increased interactions with AKT1 expressing cells compared to control cells. In vivo time-lapse imaging was performed using the mpeg1:EGFP transgenic line to observe microglia behaviour towards control cells and AKT1 cells. (A) In controls, microglia were observed to behave physiologically. Cells adopted the typical ramified morphology constantly sending out branched processes to survey the microenvironment (see also *Figure 1—video 1*). (B) Following AKT1 overexpression, microglia were observed to directly interact with AKT1+ cells (see also *Figure 1—video 2*). (C) Quantification of the percentage of microglia interacting with control and AKT1 positive cells (control: 16.86 ± 1.33%, n = 20; AKT1: 41.79 ± 2.65%, n = 21). Specific microglia interactions with AKT1+ cells include (D) the wrapping of cell bodies around the oncogenic cells (see also *Figure 1— video 3*), as well as (E) two microglial cells making direct contacts with AKT1+ via their extended processes (white arrows) (see also *Figure 1—video 4*). Representative images at five dpf are shown. Images were captured using an Andor spinning disk confocal microscope with a 20x/0.75 objective. Image acquisition was carried out over a duration of 180 min (3 hr). Scale bars represent 30 μm. Error bars represent mean ± SEM.
DOI: https://doi.org/10.7554/eLife.46912.002

The following video, source data, and figure supplement are available for figure 1:

**Source data 1.** Quantfications of microglial interactions with control and AKT1+ cells.
DOI: https://doi.org/10.7554/eLife.46912.004
**Figure supplement 1.** Microglial responses to oncogenic cells.
*Figure 1 continued on next page*

*Figure 1 continued*

DOI: https://doi.org/10.7554/eLife.46912.003

**Figure 1—video 1.** Microglia responses to control RFP neural cells (REF to *Figure 1*).

DOI: https://doi.org/10.7554/eLife.46912.005

**Figure 1—video 2.** Microglia display close interactions with AKT1 expressing cells (REF to *Figure 1*).

DOI: https://doi.org/10.7554/eLife.46912.006

**Figure 1—video 3.** Microglia display close interactions with AKT1 expressing cells (REF to *Figure 1*).

DOI: https://doi.org/10.7554/eLife.46912.007

**Figure 1—video 4.** Different microglia interact with the same isolated AKT1 expressing cell (REF to *Figure 1*).

DOI: https://doi.org/10.7554/eLife.46912.008

## AKT1-positive cells show increased intracellular $Ca^{2+}$ levels

Increased $Ca^{2+}$ levels in neurons have been shown to mediate ATP release, which stimulates microglia processes towards these neurons (*Li et al., 2012*; *Sieger et al., 2012*; *Eyo et al., 2014*; *Eyo et al., 2015*). Thus, we hypothesised that AKT1+ cells would exhibit increased intracellular $Ca^{2+}$ levels compared to control cells. To prove this hypothesis, we made use of transgenic b-actin: GCaMP6f zebrafish which ubiquitously express the calcium sensor GCamP6f. We overexpressed AKT1 in b-actin:GCaMP6F larvae and imaged the larval brains. We then quantified GCaMP6F fluorescence in AKT1+ cells compared to control cells by measuring the mean relative fluorescence intensity change ($\Delta F/F_0$) (*Baraban et al., 2018*). Indeed, these quantifications showed a steady increase of $Ca^{2+}$ levels in AKT1+ cells compared to control cells over time (*Figure 2A–C*). $Ca^{2+}$ levels were significantly increased in AKT1+ cells from four dpf onwards and showed a drastic increase at 7 dpf (*Figure 2C*). When normalised against control $\Delta F/F_0$ values, the significance was further pronounced with percentage fold change of $Ca^{2+}$ levels of AKT1 cells increasing from $189.7 \pm 70.6\%$ at 4 dpf, to $204.8 \pm 102.1\%$ (5 dpf) and $250.2 \pm 67.1\%$ (6 dpf), respectively, to over $1615.3 \pm 271.4\%$ by seven dpf. We speculate that increased $Ca^{2+}$ levels are part of the process of oncogenic transformation as we observed similar increases upon overexpression of HRASV12 in neural cells (not shown).

To test if these increased $Ca^{2+}$ levels were dynamic over time, we recorded individual brains using spinning disk confocal microscopy with a time resolution of 1 frame/s. For the analysis, the $\Delta F/F_0$ of any selected cell-of-interest was measured along the time-course and plotted as a function of $\Delta F/F_0$ against time. Interestingly, these recordings showed further differences between control cells and AKT1+ cells. In control RFP cells, calcium activity was observed to be relatively static over time (n = 35 larvae analysed; *Figure 2D*, *Figure 2—video 1*). With the exception of some spontaneous background firing, there were no spikes or obvious changes in calcium firing pattern recorded in control neural cells throughout the duration of image acquisition (*Figure 2D*, *Figure 2—video 1*). Interestingly, AKT1 expressing cells were observed to temporally regulate calcium activity. Through the course of acquisition,~31% of AKT1-positive cells were observed to strongly up- and down-regulate $Ca^{2+}$ levels ($\Delta F/F_0$ increase >0.05) repeatedly, thus creating a firing pattern (n = 35 larvae analysed; *Figure 2E*, *Figure 2—video 2*). We did not detect these patterns in any of the analysed control RFP cells (n = 35 larvae analysed). While these dynamic changes in calcium firing were specific to individual AKT1 positive cells, they were more frequently observed in cells within close vicinity to other AKT1 expressing cells. Thus, AKT1 induced pre-neoplastic alterations result in increased $Ca^{2+}$ levels, which appear to be dynamically regulated over time. To test if microglia directly respond to increases in $Ca^{2+}$ levels we overexpressed AKT1 in b-actin:GCaMP6f/mpeg1:EGFP double transgenic larvae. This combination has the caveat that microglia and $Ca^{2+}$ signals are imaged in the same channel. Thus, imaging settings had to be carefully adjusted to avoid massive overexposure of the microglia while still capturing the GCaMP6f signals. Importantly, we detected microglia directly responding to AKT1+ cells with increased $Ca^{2+}$ levels. We observed prolonged cellular contacts between microglia and AKT1+ cells with increased $Ca^{2+}$ levels (*Figure 3A–D*, arrows) as well as microglia sending processes towards AKT1+ cells that increased their $Ca^{2+}$ levels during the time of the acquisition (*Figure 3A–H*, arrowheads). These results suggest, that increased $Ca^{2+}$ levels in AKT1+ cells stimulate microglial contacts.

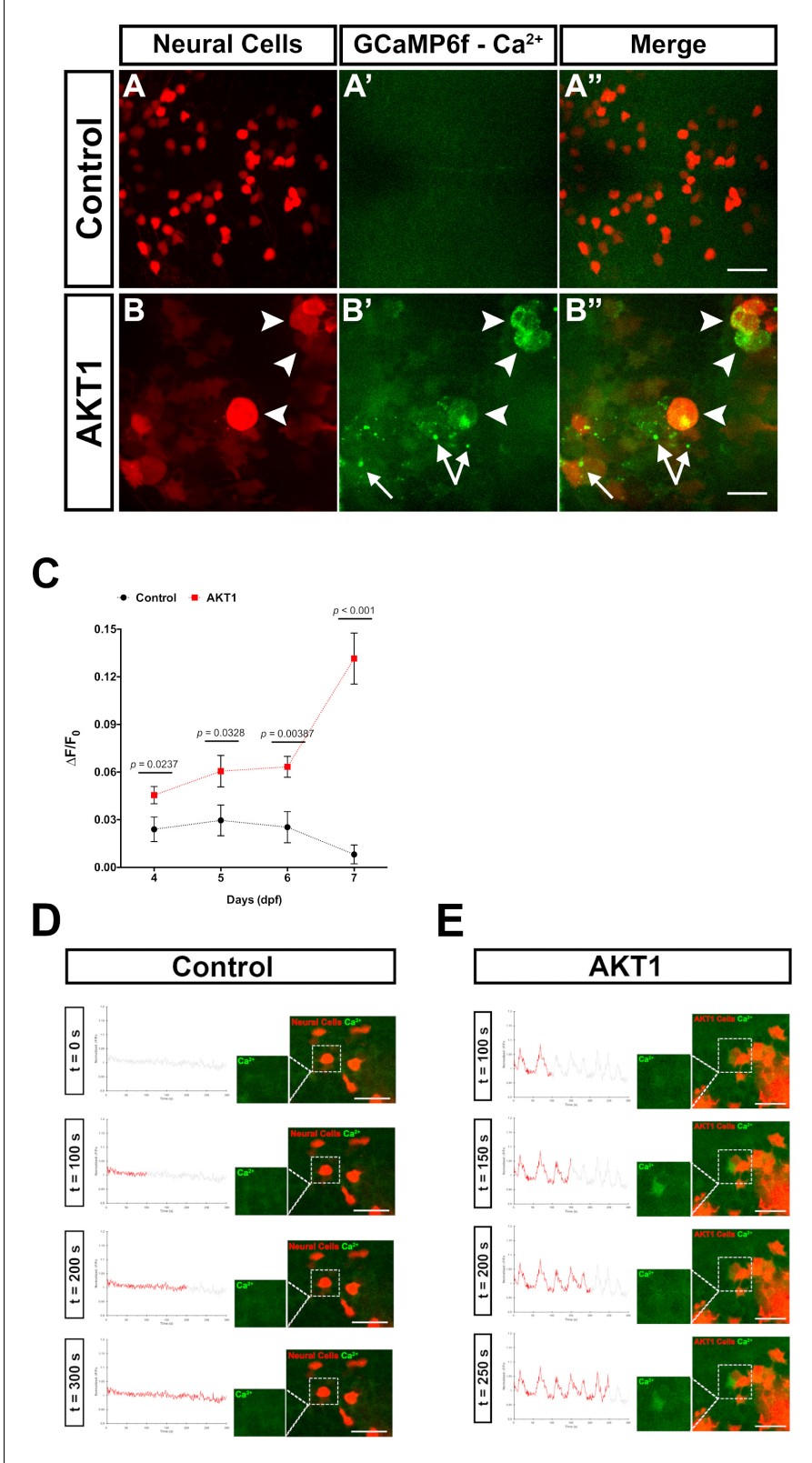

**Figure 2.** AKT1 expressing cells have increased levels of intracellular Ca$^{2+}$. The β-actin:GCaMP6f transgenic line was used to monitor and measure in vivo calcium (Ca$^{2+}$) levels in control and AKT1+ cells. (**A-A'**) Control neural cells showed a low, homogenous basal level of intracellular Ca$^{2+}$. (**B-B'**) AKT1 + cells showed cell specific increase in intracellular Ca$^{2+}$ levels (white arrowheads). (**C**) Quantification of the mean relative fluorescence intensity change (ΔF/F$_0$) of control and AKT1+ cells at 4 dpf, 5 dpf, 6 dpf, and 7 dpf. Significant differences were observed between control and AKT1 expressing larvae

*Figure 2 continued on next page*

*Figure 2 continued*

at all four time points. (Control – 4 dpf: 0.0240 ± 0.0078, n = 22; 5 dpf: 0.0296 ± 0.0097, n = 19; 6 dpf: 0.0253 ± 0.0098, n = 25; 7 dpf: 0.00815 ± 0.0059, n = 25). (AKT1 – 4 dpf: 0.0455 ± 0.0055, n = 29; 5 dpf: 0.0606 ± 0.0099, n = 22; 6 dpf: 0.0633 ± 0.0066, n = 20; 7 dpf: 0.132 ± 0.016, n = 32). Representative images of larvae at 8 dpf are shown. (D)(D) + (E) To monitor changes in $Ca^{2+}$ levels over time, samples were imaged over 5 min (300 s) with a capture rate of 1 frame/s. The data has been normalised and represented as a function of $\Delta F/F_0$ plotted against time. (D) Calcium activity in control cells showed no changes over time (n = 35 larvae analysed) (see also *Figure 2—video 1*). (E) AKT1 expressing cells were found to temporally regulate calcium activity, through up- and down-regulation of $Ca^{2+}$ levels (n = 35 larvae analysed) (see also *Figure 2—video 2*). Images were captured using an Andor spinning disk confocal microscope with a 20x/0.75 objective. Scale bars represent 20 µm. Error bars represent mean ± SEM.
DOI: https://doi.org/10.7554/eLife.46912.009

The following video and source data are available for figure 2:

**Source data 1.** Quantifications of GCaMP6F fluorescence in control and AKT1+ cells.
DOI: https://doi.org/10.7554/eLife.46912.010
**Figure 2—video 1.** Control cells show minor changes in intracellular $Ca^{2+}$levels over time (REF to *Figure 2*).
DOI: https://doi.org/10.7554/eLife.46912.012
**Figure 2—video 2.** AKT1 cells dynamically regulate their intracellular $Ca^{2+}$levels over time (REF to *Figure 2*).
DOI: https://doi.org/10.7554/eLife.46912.013

## $Ca^{2+}$-ATP-P2ry12 signalling is required for cellular contacts between microglia and AKT1+ cells

To test if increased $Ca^{2+}$ levels correlate with increased microglial interactions we took pharmacological and genetic approaches. First, we incubated larvae with a mixture of MK801 and MK5 to inhibit NMDA receptor mediated $Ca^{2+}$ entry into cells (*Sieger et al., 2012*). Inhibition of NMDA receptor signalling led to a significant reduction of $Ca^{2+}$ levels in AKT1+ cells compared to the AKT1 + cells in untreated larvae (*Figure 4A*). In line with the reduction of $Ca^{2+}$ levels we detected a significant reduction in the number of microglial interactions with AKT1+ cells (*Figure 4B*). Thus, increased $Ca^{2+}$ levels in AKT1+ cells are required to attract microglial processes.

Attraction of microglial processes to neurons with increased $Ca^{2+}$ levels have been shown to be regulated via the release of ATP/ADP, which is sensed by the P2y12 receptor expressed on microglia (*Li et al., 2012*; *Sieger et al., 2012*; *Eyo et al., 2014*; *Eyo et al., 2015*). Consequently, inhibiting ATP and P2ry12 signalling abolishes microglial responses to cellular increases in $Ca^{2+}$ levels. To test if microglial responses to AKT1+ cells were mediated via the same mechanism we reduced ATP release by treating larvae with CBX to block pannexin channels as described before (*Chekeni et al., 2010*; *Sieger et al., 2012*). Indeed, inhibiting pannexin channels led to a significant reduction of cellular interactions between microglial cells and AKT1+ cells (*Figure 4C*). Finally, we decided to inhibit P2ry12 signalling using a genetic approach. Importantly, *p2ry12* expression is highly specific to microglia in the brain and *p2ry12* is considered to be a microglia signature gene (*Crotti and Ransohoff, 2016*). Thus, we performed CRISPR manipulation with a *p2ry12* gene-specific guide RNA (gRNA). Acute injection of the *p2ry12* gRNA efficiently mutated the *p2ry12* gene as shown by restriction fragment length polymorphism (RFLP) analysis, while injection of a control gRNA did not cause mutation of the *p2ry12* gene (*Figure 4—figure supplement 1A,B*). RFLP analysis demonstrated the *p2ry12* gRNA had a mutation rate approaching 100% (*Figure 4—figure supplement 1A, B*). To further confirm efficiency on protein level, we injected gRNA into double transgenic p2ry12-GFP/mpeg1:mCherry zebrafish in which microglia (and all other macrophages) are labelled with mCherry and microglia are additionally labelled by the P2ry12-GFP fusion protein (*Figure 4—figure supplement 1C*). Importantly, the p2ry12-GFP zebrafish were created by BAC mediated recombination of a GFP fusion into the genomic *p2ry12* locus (*Sieger et al., 2012*), thus allowing assessment of endogenous P2ry12 expression. Injection of a control gRNA into these double transgenic fish did neither alter mCherry nor P2ry12-GFP expression (*Figure 4—figure supplement 1C*). Injection of *p2ry12* gRNA into these fish did not, as expected, impact on mCherry expression on microglia (*Figure 4—figure supplement 1C*). However, P2ry12-GFP expression was clearly abolished on the microglia, revealing complete knockout of the P2ry12 protein (*Figure 4—figure supplement 1C*). Thus, the gRNA injected produced an effective mosaic null, herein referred to as p2yr12 crispant. We then quantified microglial interactions with AKT1+ cells in *p2yr12* wildtype brains (no gRNA + control gRNA) and *p2ry12* crispant brains. Importantly, in the *p2ry12* crispant background microglia interactions with AKT1+ cells were significantly reduced (*Figure 4D*). Quantifications

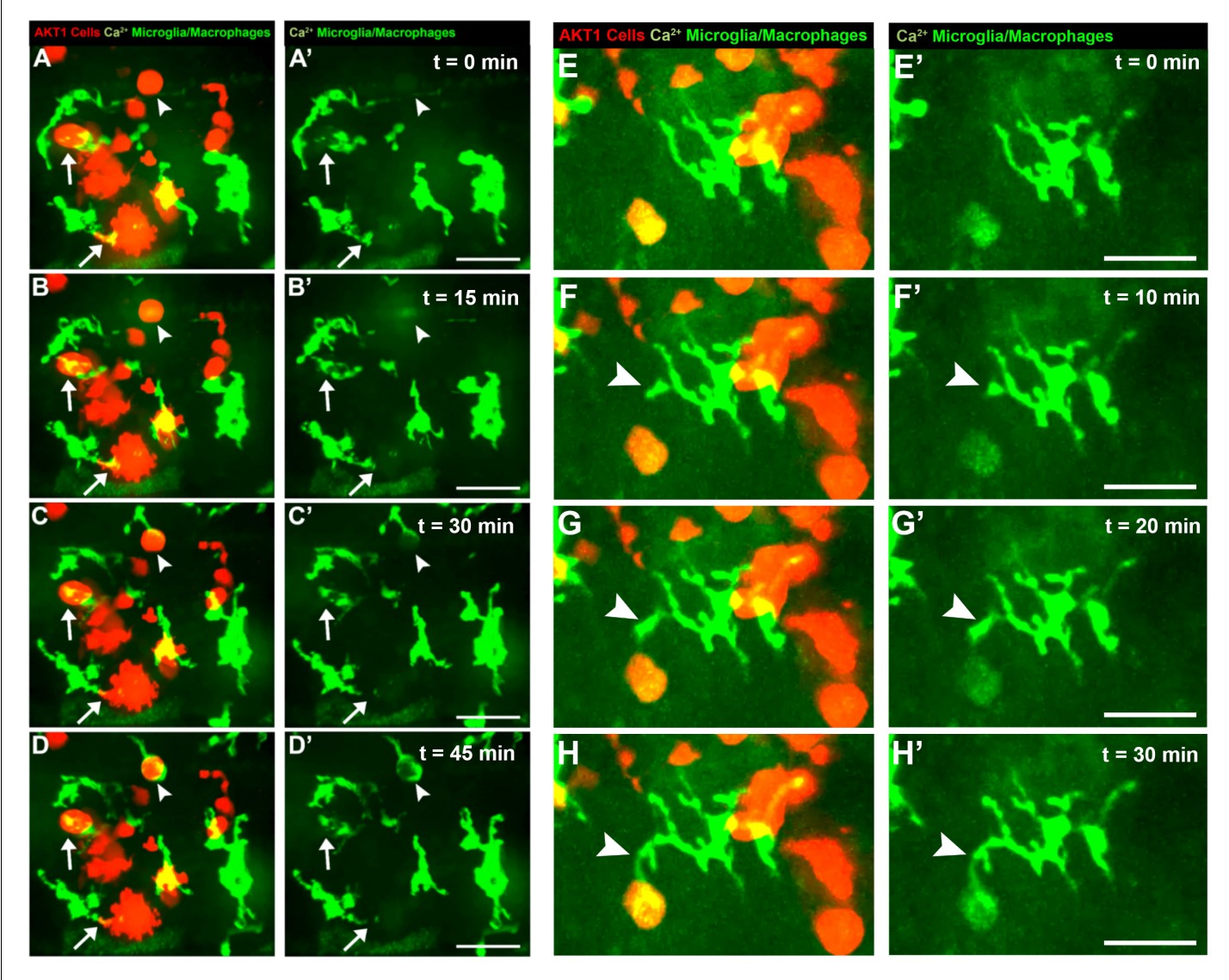

**Figure 3.** Microglia directly respond to increased levels of intracellular Ca$^{2+}$ in AKT1+ cells. Microglia were observed to display various different responses towards AKT1 positive cells with upregulated Ca$^{2+}$ levels. One type of interaction was the prolonged cell-to-cell contact between the microglial cell and the AKT1 expressing cell (A-D, arrows). In addition, microglia were observed to extend processes towards AKT1 cells with increased calcium activities (A-H, arrowheads). Representative images at five dpf are shown. Images were captured using an Andor spinning disk confocal microscope with a 20x/0.75 objective. Scale bars represent 20 μm.

DOI: https://doi.org/10.7554/eLife.46912.011

revealed that while on average ~40% of microglia interacted with AKT1+ cells in *p2ry12* wt brains, only ~21% of microglia showed interactions with AKT1+ cells in the *p2ry12* crispant background (*Figure 4D*).

In conclusion, these experiments show that P2ry12 signalling in microglia is required to mediate cellular interactions with AKT1+ cells.

## Microglial interactions promote proliferation of AKT1+ cells

We have shown that microglial interactions with AKT1+ cells were abolished in *p2ry12* crispant brains. To test if the reduced number of interactions had a direct functional impact on the growth of AKT1+ cells, we measured proliferation rates of AKT1+ cells in *p2ry12* wildtype brains (no gRNA + control gRNA) and *p2ry12* crispant brains. As described previously, AKT1+ cells showed

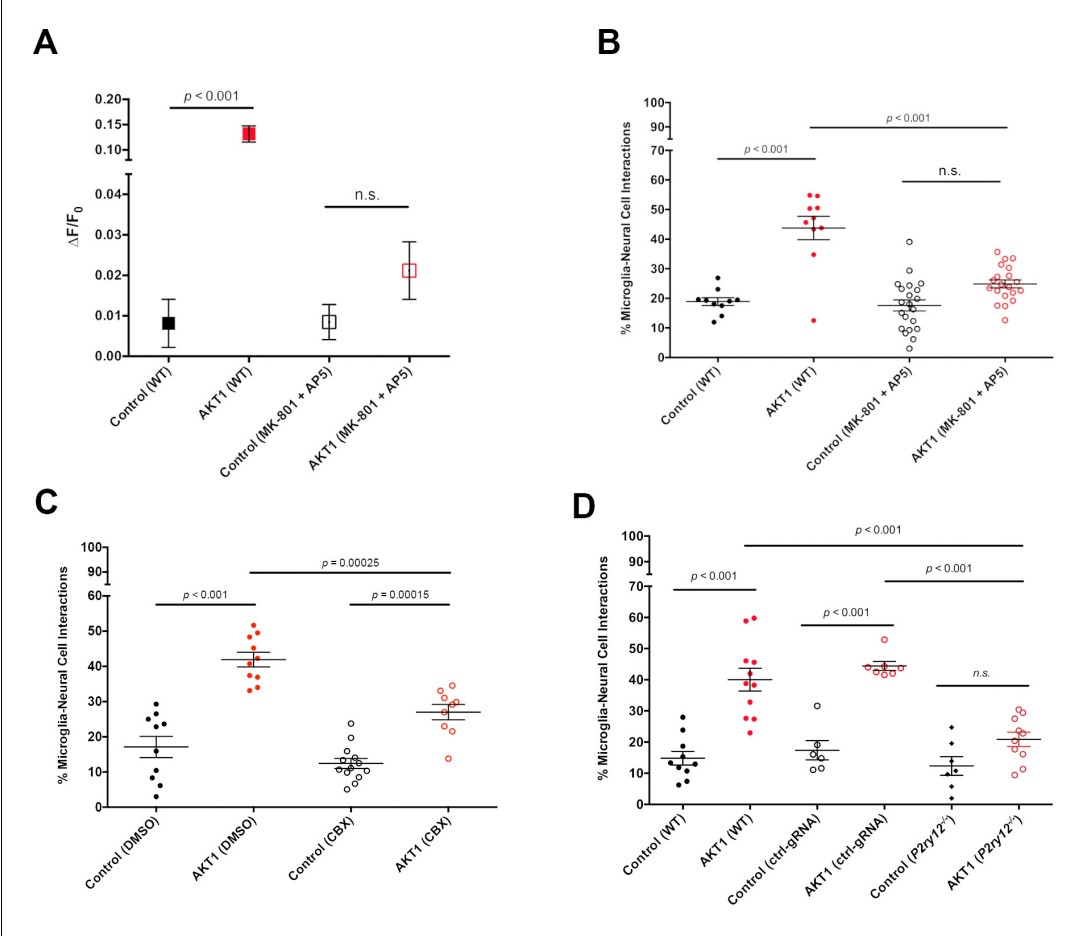

**Figure 4.** $Ca^{2+}$-ATP-P2ry12 signalling stimulates microglial interactions with AKT1 cells. The β-actin:GCaMP6f transgenic line was used to monitor and measure in vivo calcium ($Ca^{2+}$) levels in control and AKT1 expressing cells. The mpeg1:EGFP transgenic line was used to quantify microglial interactions with control and AKT1 cells. (**A**) Treating larvae with MK801 and MK5 to inhibit NMDA receptor signalling led to a significant reduction of $Ca^{2+}$ levels in treated AKT1 cells compared to untreated AKT1 cells. Quantification of the mean relative fluorescence intensity ($\Delta F/F_0$) of $Ca^{2+}$ levels in control and in AKT1 expressing cells is shown (control (WT): 0.0081 ± 0.006, n = 25; AKT1 (WT): 0.1316 ± 0.016, n = 32; control (MK801 +MK5): 0.0085 ± 0.004, n = 16; AKT1 (MK801 +MK5): 0.0211 ± 0.007, n = 16). (**B**) The percentage of microglial cells interacting with AKT1 cells was significantly reduced in larvae treated with MK801 and MK5 compared to untreated larvae. (Control (WT): 18.89 ± 1.32, n = 10; AKT1 (WT): 43.75 ± 3.95, n = 10; Control (MK801 +MK5): 17.59 ± 1.89, n = 21; AKT1 (MK801 +MK5): 24.94 ± 1.36, n = 20). (**C**) The percentage of microglial cells interacting with AKT1 cells was significantly reduced in larvae treated with CBX compared to untreated larvae (Control (DMSO): 17.11 ± 3.02%, n = 10; AKT1 (DMSO): 41.92 ± 2.09%, n = 10; Control (CBX): 12.42 ± 1.42%, n = 13; AKT1 (CBX): 26.99 ± 2.19%, n = 9).(**D**) The percentage of microglial cells interacting with AKT1 cells was significantly reduced in *p2ry12* crispant larvae compared to WT larvae (Control (WT): 14.82 ± 2.19%, n = 10; AKT1 (WT): 40.01 ± 3.66%, n = 11; Control (ctrl-gRNA): 17.38 ± 3.09%, n = 6; AKT1 (ctrl-gRNA): 44.42 ± 1.46%, n = 7; Control (*p2ry12*[-/-]): 12.33 ± 2.97%, n = 7; AKT1 (*p2ry12*[-/-]): 20.88 ± 2.29%, n = 10).

DOI: https://doi.org/10.7554/eLife.46912.014

The following source data and figure supplement are available for figure 4:

**Source data 1.** Quantfications of microglial interactions with control and AKT1+ cells upon interference with Ca2+ - ATP -P2ry12 signalling..
DOI: https://doi.org/10.7554/eLife.46912.016
**Figure supplement 1.** CRISPR/Cas9-mediated mutation of the *p2ry12* gene.
DOI: https://doi.org/10.7554/eLife.46912.015

significantly increased proliferation rates compared to control cells in wt brains (*Figure 5B*) (*Chia et al., 2018*). In *p2ry12* crispant brains no differences were detected in the proliferation rates of control RFP cells compared to *p2yr12* wildtype brains (*Figure 5B*). However, we found an almost 50% drop in the proliferation rate of AKT1+ cells in *p2ry12* crispant brains compared to *p2ry12* wild-type brains (*Figure 5B*). As numbers of microglia were similar in *p2ry12* crispant brains and *p2yr12* wildtype brains (*Figure 5A*), we conclude that the reduced number of interactions in *p2ry12* crispant

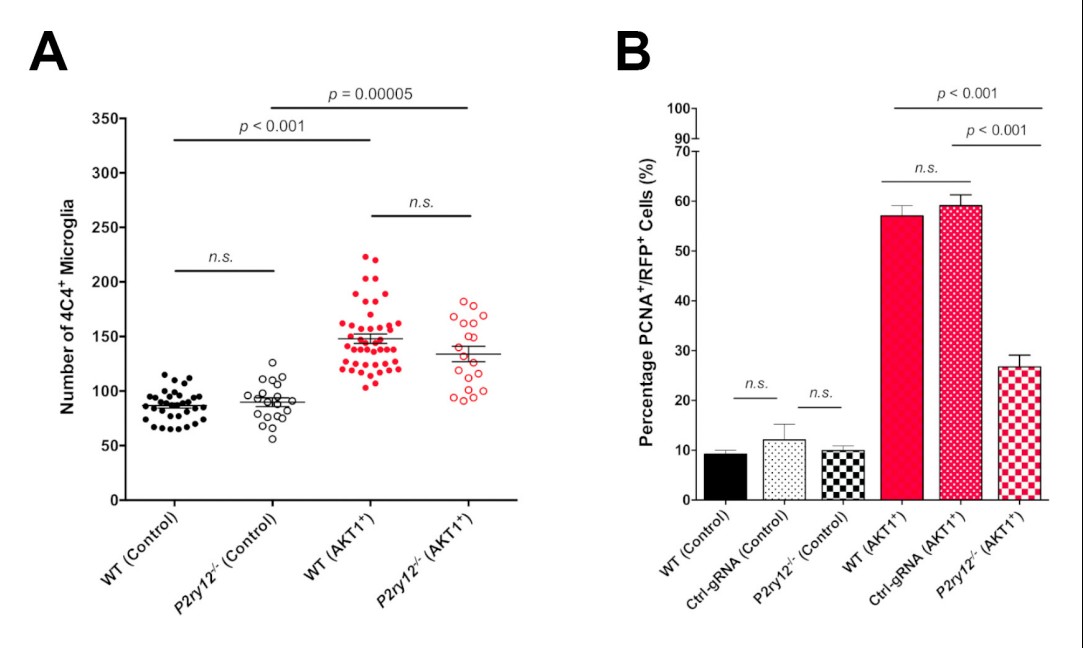

**Figure 5.** P2RY12-mediated microglial interactions stimulate AKT1 cell proliferation. CRISPR/Cas9-mediated knockout of the P2y12 receptor had no impact on microglia numbers but led to significantly reduced proliferation rates of AKT1+ cells. (**A**) Quantification of the number of microglia in control larvae and upon AKT1 overexpression in WT and *p2ry12* crispant zebrafish (Control – WT: 86.71 ± 2.34, n = 34; *p2ry12*$^{-/-}$: 89.8 ± 3.99, n = 20) (AKT1 – WT: 148 ± 4.38, n = 45; *p2ry12*$^{-/-}$: 133.9 ± 7.07, n = 19). (**B**) Quantification of the level of proliferation of RFP-expressing cells in control larvae and upon AKT1 overexpression in WT, ctrl-gRNA and *p2ry12* crispant zebrafish (Control – WT: 9.25 ± 0.75%, n = 13; ctrl-gRNA: 12.07 ± 3.16%, n = 11; *P2ry12*$^{-/-}$: 9.92 ± 0.97%, n = 20) (AKT1 – WT: 57.1 ± 2.03%, n = 17; ctrl-gRNA: 59.12 ± 2.18%, n = 12; *P2ry12*$^{-/-}$: 26.8 ± 2.37%, n = 19). Error bars represent mean ± SEM.

DOI: https://doi.org/10.7554/eLife.46912.017

The following source data is available for figure 5:

**Source data 1.** Quantifications of microglial numbers and proliferation of neural cells in P2ry12 crispants and controls.
DOI: https://doi.org/10.7554/eLife.46912.018

brains was the reason for the decrease in proliferation of AKT1 cells. Thus, microglia interactions directly promote proliferation of pre-neoplastic AKT1 cells.

## Discussion

A number of elegant studies have described the mechanism how microglial processes are attracted towards highly active neurons and injuries within the brain (*Davalos et al., 2005*; *Li et al., 2012*; *Sieger et al., 2012*; *Eyo et al., 2014*; *Eyo et al., 2015*). Here, we showed that pre-neoplastic cells hijack the same mechanism to attract microglial processes. We showed that increased Ca$^{2+}$ levels in AKT1+ cells, the release of ATP from these cells and P2ry12 signalling on microglia are required to stimulate microglial interactions with AKT1+ cells. Intriguingly, we showed that these interactions promote an increase in proliferation of the pre-neoplastic cells. Thus, we have identified a new process that contributes to the pro-tumoural activities of microglia.

A variety of mechanisms have been described how macrophages and microglia promote the growth of tumours. These mechanisms range from the release of cytokines and chemokines to modifications of the extracellular matrix (reviewed in *Hambardzumyan et al. (2016)*). Here, we identify direct cellular interactions between microglia and pre-neoplastic cells as a cause of increased proliferation. Interestingly, cellular interactions between macrophages and tumour cells have been described recently in other tumour contexts. Roh-Johnson et al. showed direct cellular contacts between macrophages and melanoma cells (*Roh-Johnson et al., 2017*). These cellular contacts

resulted in the transfer of cytoplasm from macrophages to melanoma cells which led to an increased dissemination of the melanoma cells (*Roh-Johnson et al., 2017*). Furthermore, macrophage contacts with breast cancer cells have been shown to induce RhoA GTPase signalling within the cancer cells and to trigger their intravasation (*Roh-Johnson et al., 2014*). Nevertheless, several open questions remain to be answered here. What is the content within the transferred cytoplasm that leads to increased invasiveness of melanoma cells? How do macrophages upregulate RhoA GTPase activity within breast cancer cells? How do microglia induce proliferation of pre-neoplastic cells? We hypothesise that microglial processes alter the $Ca^{2+}$ levels within the pre-neoplastic cells which might trigger changes in their proliferative capacities. This is in line with previous studies showing a $Ca^{2+}$-dependent increase in transcription factors which are crucial for cellular division, proliferation, as well as cancer cell survival (reviewed by *Roderick and Cook, 2008*). Future studies will reveal if the microglia-mediated increase in proliferation is mediated via ligand-receptor interactions or via a transfer of cytoplasm as shown for macrophages and melanoma cells previously.

AKT1+ cells did not only show increased $Ca^{2+}$ levels but also showed a dynamic regulation of $Ca^{2+}$ levels. Interestingly, cells that were within close vicinity to other AKT1 expressing cells showed an increased frequency of fluctuations in their $Ca^{2+}$ levels. Thus, it's tempting to speculate that these cells communicate via $Ca^{2+}$ transients. Interestingly, in an elegant in vivo study, astrocytoma cells have been shown to form a functional network which is connected via tumour microtubes (TMs) (*Osswald et al., 2015*). Astrocytoma cells within the network showed survival benefits and resistance against radiotherapy (*Osswald et al., 2015*). Importantly, dynamic $Ca^{2+}$ transients have been observed within this network and a role for the network in brain invasion and proliferation has been described. Future studies will address if the $Ca^{2+}$ transients within AKT1+ cells shown here resemble early signs of network formation. As we observed microglia to directly interact with these AKT1+ cells and their processes, we speculate that microglia contribute to the establishment of a functional network between tumour cells by promoting the outgrowth of TMs. This might be mediated by factors that have been identified before and shown to be involved in developmental processes. Microglia have been shown for example to promote neuronal survival and axonal growth by providing insulin-like growth factor 1 (IGF-1) (*Ueno et al., 2013*). Thus, it will be interesting to analyse the role of IGF-1 and other developmental factors in the outgrowth of TMs.

Future studies will reveal if the mechanism identified here is employed by the large variety of brain tumours and if promotion of proliferation by microglial processes is a general phenomenon within brain tumours. As expression of the P2y12 receptor, which mediates these interactions, is specific for microglia in the brain, pharmacological inhibition of the receptor might offer new routes for therapy to reduce proliferation of the tumour cells.

## Materials and methods

**Key resources table**

| Reagent type (species) or resource | Designation | Source or reference | Identifiers | Additional information |
|---|---|---|---|---|
| Antibody | anti-4C4 (mouse monoclonal) | Becker Lab, University of Edinburgh | | (1:50) |
| Antibody | anti-PCNA (rabbit polyclonal) | abcam | abcam: ab18197; RRID:AB_2160346 | (1:300) |
| Antibody | Alexa 488- or 647 secondaries | Life Technologies | Life Technologies: A11001 (RRID:AB_138404), A21235 (RRID:AB_141693), A11008 (RRID:AB_143165), A21244 (RRID:AB_141663) | (1:200) |
| Chemical compound, drug | Carbenoxolone (CBX) | Sigma-Aldrich | Sigma-Aldrich: C 4790 | 50 µM, 1% DMSO |
| Chemical compound, drug | MK-801 | Sigma-Aldrich | Sigma-Aldrich: M107 | 100 µM |

*Continued on next page*

*Continued*

| Reagent type (species) or resource | Designation | Source or reference | Identifiers | Additional information |
|---|---|---|---|---|
| Chemical compound, drug | AP5 | Sigma-Aldrich | Sigma-Aldrich: A5282 | 10 µM |
| Gene (*Homo sapiens*) | AKT1 | NA | ENSG00000142208 | |
| Gene (*Homo sapiens*) | HRASV12 | NA | ENSG00000174775 | |
| Recombinant DNA reagent | lexOP-AKT1-RFP (plasmid) | *Chia et al., 2018* | lexOP:AKT1-lexOP:tagRFP | Gateway vector: pDEST |
| Recombinant DNA reagent | lexOP-HRASV12-RFP (plasmid) | this paper | lexOP:HRASV12-lexOP:tagRFP | Gateway vector: pDEST |
| Recombinant DNA reagent | UAS-AKT1-BFP (plasmid) | this paper | UAS:AKT1:UAS:BFP | Gateway vector: pDEST |
| Recombinant DNA reagent | UAS-eGFP-HRASV12 (plasmid) | PMID: 27935819 | UAS:EGFP-HRASV12 | Gateway vector: pDEST |
| Recombinant DNA reagent | lexOP-tagRFP (plasmid) | *Chia et al., 2018* | lexOP:tagRFP-pA | Gateway vector: pDEST |
| Strain, strain background (*D. rerio*) | zic:Gal4 | *Distel et al., 2009* | Et(zic4:GAL4TA4,UAS:mCherry)hmz5, ZDB-ETCONSTRCT-110214–1 | |
| Strain, strain background (*D. rerio*) | b-actin:GCaMP6f | *Herzog et al., 2019* | Tg(b-actin:GCaMP6f) | |
| Strain, strain background (*D. rerio*) | mpeg1:EGFP | *Ellett et al., 2011* | Tg(mpeg1:EGFP)gl22, RRID:ZIRC_ZL9940 | |
| Strain, strain background (*D. rerio*) | mpeg1:mCherry | *Ellett et al., 2011* | Tg(mpeg1:mCherry)gl23, RRID:ZIRC_ZL9939 | |
| Strain, strain background (*D. rerio*) | NBT:ΔlexPR-lexOP-pA | *Chia et al., 2018* | Tg(Xla.Tubb:LEXPR)Ed7, ZDB-ALT-180108–4 | |
| Strain, strain background (*D. rerio*) | p2ry12:p2ry12-GFP | *Sieger et al., 2012* | TgBAC(p2ry12:p2ry12-GFP), RRID:ZFIN_ZDB-ALT-121109-2 | |
| Software, algorithm | Imaris 8.0.2 | Bitplane | RRID:SCR_007370 | |
| Chemical compound, drug | TracrRNA | Merck | Merck: TRACRRNA05N | |
| Chemical compound, drug | guide RNA | Merck | Merck: custom made | |
| Peptide, recombinant protein | Cas9 nuclease | NEB | NEB: M0386M | |

## Zebrafish maintenance

Zebrafish were housed in a purpose-built zebrafish facility, in the Queen's Medical Research Institute, maintained by the University of Edinburgh Bioresearch and Veterinary Services. All zebrafish larvae were kept at 28°C on a 14 hr light/10 hr dark photoperiod. Embryos were obtained by natural spawning from adult Tg(mpeg1:EGFP)gl22 referred to as mpeg1:EGFP (*Ellett et al., 2011*), Tg (mpeg1:mCherry) referred to as mpeg1:mCherry, wildtype (AB), TgBAC(p2ry12:p2ry12-GFP)hdb3 referred to as p2ry12:p2ry12-GFP (*Sieger et al., 2012*), Et(zic4:GAL4TA4,UAS:mCherry)hmz5 referred to as zic4:Gal4 (*Distel et al., 2009*) and Tg(b-actin:GCaMP6f) referred to as b-actin: GCaMP6f (*Herzog et al., 2019*) and Tg(Xla.Tubb:LEXPR)Ed7 referred to as NBT:ΔlexPR-lexOP-pA (NBT:ΔlexPR) (*Chia et al., 2018*). Embryos were raised at 28.5°C in embryo medium (E3) and treated with 200 µM 1-phenyl 2-thiourea (PTU) (Sigma) from the end of the first day of development for the duration of the experiment to inhibit pigmentation. Animal experimentation was approved by the ethical review committee of the University of Edinburgh and the Home Office, in accordance with the Animal (Scientific Procedures) Act 1986.

DNA injections to induce oncogene expression and cellular transformation.

To achieve transient expression of AKT1 and HRASV12, zebrafish embryos were injected at the 1 cell stage as previously described (*Chia et al., 2018*). Approximately 2 nL of plasmid DNA (30 ng/µL) containing Tol2 capped mRNA (20 ng/µL) and 0.2% phenol red were injected into NBT:ΔlexPR-lexOP-pA fish. To obtain AKT1 or HRASV12 expression in other transgenic backgrounds, a Tol2-pDEST-NBT:ΔlexPR-lexOP-pA (20 ng/µL) plasmid was co-injected with a Tol2-pDEST-lexOP:*AKT1*-lexOP:tagRFP (30 ng/µL) plasmid or with a Tol2-pDEST-lexOP:*HRASV12*-lexOP:tagRFP (30 ng/µL) plasmid. To obtain control RFP expression Tol2-pDEST-lexOP:tagRFP-pA was injected. To obtain AKT1 expression under control of the zic4 enhancer the zic:Gal4 line was crossed with mpeg1:EGFP and injected with a Tol2-pDEST-UAS:*AKT1*-UAS:BFP (30 ng/µL) plasmid. To obtain HRASV12 expression under control of the zic4 enhancer the zic:Gal4 line was used and injected with a Tol2-pDEST-UAS:*eGFP-HRASV12* (30 ng/µL) plasmid. Larvae were screened at 2 days post-fertilisation (dpf) for positive transgene expression and selected for further experiments.

## CRISPR/Cas9-mediated *p2ry12* mutation

Somatic mosaic *p2ry12* mutations were generated via a CRISPR/Cas9 approach as described before (*Tsarouchas et al., 2018*). The CrRNA for *p2yr12* (target sequence: 5′-CCAGTTCTACTACC TGCCCACGG-3′, targeting a *bsl1* restriction enzyme site), the control CrRNA (target sequence: 5′-CCTCTTACCTCAGTTACAATTTATA-3′) and the TracrRNA were ordered from Merck KGaA (Germany, Darmstadt). The injection mix included 1 µl TracrRNA 250 ng/µl, 1 µl CrRNA 250 ng/ul, 1 µl Cas9 protein 1 µM (NEB). To knock out *p2ry12* and obtain AKT1 expression in the same larva, a Tol2-pDEST-lexOP:*AKT1*-lexOP:tagRFP was co-injected with the CRISPR/Cas9 injection mix of Cas9 protein, TracrRNA, and *p2ry12 CrRNA* or control CrRNA. To obtain experimental controls, the Tol2-pDEST-lexOP:tagRFP-pA was co-injected. To confirm p2yr12 locus had been mutated restriction fragment length polymorphism (RFLP) analysis was performed using the *bsl1* enzyme (NEB). The PCR primer pair used was: Forward primer: 5′-AGCTCAGCTTCTCCAACAGC-3′; Reverse primer: 5′GCTACATTGGCAT CGGATAA-3′. PCR products were digested with the *bsl1* restriction enzyme (55°C for 1 hr).

## Whole mount immunohistochemistry, image acquisition and live imaging

Whole mount immunostaining of samples was performed as previously describe (*Chia et al., 2018*). Briefly, larvae were fixed in 4% PFA/1% DMSO at room temperature for 2 hr, followed by a number of washes in PBStx (0.2% Triton X-100 in 0.01 M PBS), and blocked in 1% blocking buffer (1% normal goat serum, 1% DMSO, 1% BSA, 0.7% Triton X-100 in 0.01 M PBS) for 2 hr prior to incubation with primary antibodies overnight at 4°C. Primary antibodies used were rabbit anti-PCNA (1:300) (ab18197, abcam) and mouse anti-4C4 (1:50). A series of washes in PBStx was carried out before samples were subsequently incubated in conjugated secondary antibodies (goat anti-mouse Alexa Fluor 488 [1:200]; goat anti-mouse Alexa Fluor 647 [1:200]; goat anti-rabbit Alexa Fluor 488 [1:200]; goat anti-rabbit Alexa Fluor 647 [1:200]) (Life Technologies) overnight at 4°C to reveal primary antibody localizations. Samples were washed following secondary antibody incubation and kept in 70% glycerol at 4°C until final mounting in 1.5% low melting point agarose (Life Technologies) in E3 for image acquisition.

Whole brain immuno-fluorescent images were acquired using confocal laser scanning microscopy (Zeiss LSM710 and LSM780; 20x/0.8 objective; 2.30 µm intervals; 488-, 543-, and 633 nm laser lines).

Live imaging of zebrafish larvae was performed as previously described (*Chia et al., 2018*); samples were anaesthetized with 0.2 mg/mL Tricaine (MS222, Sigma) and mounted dorsal side down in 1.5% low melting point agarose (Life Technologies), in glass-bottom dishes (MatTek) filled with E3 containing 0.2 mg/mL Tricaine. Single time-point live images were acquired through confocal imaging (Zeiss LSM710; 20x/0.8 objective; 2.30 µm intervals; 488-, and 543 nm laser lines). To investigate GCamp6f fluorescence and direct interactions between oncogene-expressing cells and microglia, time-lapse imaging was performed on a spinning disk confocal microscope (Andor iQ3; 20x/0.75 and W40x/1.15 objectives; 1.5–2 µm z-intervals; 488- and 543 nm laser lines). All time-lapse acquisitions were carried out in temperature-controlled climate chambers set to 28°C for 10–18 hr.

## Image analysis and quantifications

Analyses of all images were conducted using Imaris (Bitplane, Zurich, Switzerland). For the quantification of $4C4^+$ cells, only cells within the brain (telencephalon, tectum, and cerebellum) were counted for each sample using the 'Spots' function tool in Imaris 8.0.2.

To quantify proliferation rates, the number of $PCNA^+/RFP^+$ cells were counted in relation to the total number of $RFP^+$ cells and the averaged value expressed as measure of percentage proliferation (described before in *Chia et al., 2018*).

To quantify the relative $Ca^{2+}$ (GCaMP6f) intensity levels used for analyses, the mean relative fluorescence intensity change ($\Delta F/F_0$) was determined. The mean intensity of the GCaMP6f channel from the image was used to acquire the $Ca^{2+}$ baseline fluorescence intensity ($F_0$). To determine the fluorescence intensity change ($\Delta F$), the 'Surfaces' function in Imaris 8.0.2 was utilized to identify and segment all RFP+ neural cells. The mean $Ca^{2+}$ (GCaMP6f) intensity levels were recorded for individual cells. The final $\Delta F$ used was determined as the average mean intensity of the GCaMP6f fluorescence from all the segmented cells-of-interest.

Changes in $Ca^{2+}$ levels over time were analysed as previously described (*Baraban et al., 2018*). Briefly, to quantify changes in calcium activity over time in individual cells, regions of interest (ROI) were manually applied to identify the cell-of-interest. A separate ROI was applied to an area with no GCaMP6f expression to represent background fluorescence. To determine the final $\Delta F/F_0$, the following formula was applied: $\Delta F/F_0 = (F_t - F_0)/(F_0 - F_{background})$; where $F_t$ is the fluorescent intensity in the ROI at time-point '*t*', $F_0$ represents the average fluorescent intensity of the first 10 frames of the ROI, and $F_{background}$ represents the fluorescent intensity of the background ROI at time-point '*t*'.

To observe microglia interactions with RFP control cells and AKT1 positive cells, control RFP or AKT1 expression was induced in the mpeg1:EGFP transgenic zebrafish line. To determine microglial-neural cell interactions, the number of mpeg+ microglia cells in direct contact with control RFP cells or AKT1 cells were counted. To normalise for the difference in microglial numbers across samples this number was divided by the total number of mpeg+ microglial cells of the respective sample. Quantifications were plotted as the percentage of microglial cells in contact with RFP or AKT1 cells.

## Pharmacological treatments

To inhibit ATP release larvae were treated with CBX 50 µM/1% DMSO (Sigma) from 3 dpf until five dpf. To obtain experimental controls, age-matched samples were incubated in 1% DMSO. CBX treated larvae appeared inactive compared to controls and showed a reduced escape reaction upon mechanical stimulation. Inhibition of NMDA receptor signalling was achieved by treating larvae with a mixture of MK801 (100 µM) and AP5 (10 µM) (both Sigma) at 7 dpf for 5 hr.

## Statistical analysis

All experiments were performed in at least two replicates with n indicating the total number of larvae. All measured data were analyzed (StatPlus, AnalystSoft Inc). Two-tailed Student's *t*-tests were performed between two experimental groups and one-way ANOVA with Bonferroni's post-hoc tests or two-way ANOVA were performed for comparisons between multiple experimental groups. Statistical values of $p < 0.05$ were considered to be significant. All graphs were plotted in Prism 6.1 (GraphPad Software) and values presented as population means ($\pm$ SEM).

## Acknowledgements

The authors thank the BRR zebrafish facility (QMRI, University of Edinburgh) for maintenance and care of the zebrafish. The authors are grateful to members of the CALM and SURF facilities (University of Edinburgh) for assistance with microscope imaging. The authors are grateful to Graham Lieschke for sharing mpeg1:EGFP and mpeg1:mCherry fish. Thanks to Francesca Peri for sharing b-actin:GCaMP6f zebrafish. We thank Katy Astell for critical reading of this article. DS was supported by a Cancer Research UK Career Establishment Award.

## Additional information

### Funding

| Funder | Grant reference number | Author |
| --- | --- | --- |
| Cancer Research UK | C49916/A17494 | Dirk Sieger |

The funders had no role in study design, data collection and interpretation, or the decision to submit the work for publication.

### Author contributions

Kelda Chia, Formal analysis, Investigation, Visualization, Methodology, Writing—review and editing; Marcus Keatinge, Investigation, Methodology, Writing—review and editing; Julie Mazzolini, Investigation, Methodology; Dirk Sieger, Conceptualization, Resources, Data curation, Supervision, Funding acquisition, Investigation, Methodology, Writing—original draft, Writing—review and editing

### Author ORCIDs

Dirk Sieger https://orcid.org/0000-0001-6881-5183

### Ethics

Animal experimentation: Animal experimentation was reviewed and approved by the ethical review committee of the University of Edinburgh and the Home Office (Project license 60/4544 + P5042DEFB), in accordance with the Scientific Procedure Act 1986.

### Decision letter and Author response

Decision letter https://doi.org/10.7554/eLife.46912.022
Author response https://doi.org/10.7554/eLife.46912.023

## Additional files

### Supplementary files

• Transparent reporting form
DOI: https://doi.org/10.7554/eLife.46912.019

### Data availability

All data generated or analysed during this study are included in the manuscript and supporting files. Source data files have been provided for Figures 1, 2, 4 and 5.

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
