## [Decision Letter]

Thank you for submitting your article "Brain tumours repurpose endogenous neuron to microglia signalling mechanisms to promote their own proliferation" for consideration by *eLife*. Your article has been reviewed by two peer reviewers, and the evaluation has been overseen by a Reviewing Editor and Tadatsugu Taniguchi as the Senior Editor. The following individuals involved in review of your submission have agreed to reveal their identity: Jean-Pierre Levraud (Reviewer #2).

The reviewers have discussed the reviews with one another and the Reviewing Editor has drafted this decision to help you prepare a revised submission.

In this follow-up study to their previous paper, the authors now elucidate a new mechanism for the observe interaction between tumor cells and microglia. As you will see in the individual reviews below, the reviewers are generally enthusiastic about the fundamental observations but have several concerns. Specifically, better quantification to rule out that some of the observed interactions are not simply due to crowding would be important.

Summary:

In this direct follow-up of the manuscript published in *eLife* one year ago, Chia et al. deepen the analysis of the interactions of microglia with pre-tumoral cells in their model of mosaic overexpression of AKT1 in zebrafish neural cells. This time, they show that tumoral neural cells have elevated Ca^2+^ levels and sometimes pulse, and they propose that, as do active neurons, this attracts microglia via ATP secretion, and induces direct interactions that somehow promote tumor proliferation. In support of their hypothesis, they show that microglia have more frequent contacts with pretumoral cells that control cells and that this is abrogated by Ca^2+^ entry blockade. Furthermore, using CRISPR-mediated deletion of the gene encoding the p2ry12, a purinergic receptor expressed by microglia, they find that both frequency of contacts and proliferation of tumor cells depend on this receptor function.

Essential revisions:

1) The efficiency of CRISPR-mediated gene deletion in F_0_ fish is impressive, but the controls are improper. Controls should receive CAS9, the TracrRNA and a control guide RNA.

2) A central tenet of this work is that tumor cells attract microglia at short range. However, the higher frequency or contact of microglia with tumor cells (Figure 1B) could be trivially explained by the larger number of microglial cells (previous paper, Figure 4A) as well as by the larger number of tumor cells (and perhaps also their larger size, see Figure 1A) compared to control cells. Assuming random movements of microglia, frequency of interactions should be, in first approximation, proportional to the product of the frequencies of each cell type. If the measured number of contacts are normalized by this product of cell frequencies, is there still a significant difference?

I wonder about the specificity of this interaction (although the authors do show videos of the microglia interacting with/directly touching the neural cells) – could they show another cell type in the brain that isn't contacted in the same way as the neural cells?

3) The authors claim that AKT1+ cells display more frequent Ca^2+^ firing and show nice examples (2E, 2F) but a quantification should be provided.

4) Are tumor cells with the highest Ca^2+^ levels more frequently in contact with microglia? This would certainly reinforce the author's claims. This may not be trivial as the Ca^2+^ reporter use the same fluorescence channel as the macrophage reporter, but the patterns are very different so it should be doable.

5) It is satisfying to draw a straight line from AKT1 activation to Ca^2+^ to ATP release to p2ry12 receptor. The logic seemed to be that increased calcium leads to ATP release which would link to microglial interactions. But, why would increased AKT1 activity lead to increased intracellular Ca^2+^? Is this a direct effect, or are there expected to be many steps between AKT1 and Ca increase? Does AKT1 have other important effects? If Ca^2+^ is the primary mediator of the effect, can they induce increased Ca^2+^ in neurons via another mechanism (e.g. drug treatment or genetic) and see a similar effect?

6) Could loss of p2ry12 in microglia lead to other deficiencies/abnormalities in the microglia? They have similar numbers of microglia per the author's data, but do they move around as much? Do they extend the same number of processes? One might predict that increased p2ry12 expression on microglia might lead to increased interactions with neurons (even in the absence of AKT1+ transgene expression with "basal" ATP release) and then even potentially increased neural cell numbers, if this is a primary mechanism by which microglia induce neural cell number increase.

---

## [Author Response]

Essential revisions:1) The efficiency of CRISPR-mediated gene deletion in F_0_ fish is impressive, but the controls are improper. Controls should receive CAS9, the TracrRNA and a control guide RNA.

We fully agree and have now included the controls which were injected with CAS9, tracrRNA and a control guide RNA. As shown in Figure 4—figure supplement 1, injection of the control guide RNA did neither cause mutation of the p2ry12 locus nor abolish p2ry12-GFP expression in the p2ry12:p2ry12-GFP transgene. Furthermore, control guide RNA injection did not impact on interactions between microglia and AKT1 positive cells (Figure 4D) and proliferation rates of AKT1 positive cells were not altered in control guide RNA injected larvae (Figure 5B).

2) A central tenet of this work is that tumor cells attract microglia at short range. However, the higher frequency or contact of microglia with tumor cells (Figure 1B) could be trivially explained by the larger number of microglial cells (previous paper, Figure 4A) as well as by the larger number of tumor cells (and perhaps also their larger size, see Figure 1A) compared to control cells. Assuming random movements of microglia, frequency of interactions should be, in first approximation, proportional to the product of the frequencies of each cell type. If the measured number of contacts are normalized by this product of cell frequencies, is there still a significant difference?

This is a very interesting point and in first approximation contacts might be enforced by high density of cells by chance. However, we have already addressed this point in our previous paper (Chia et al., 2018) and have additional lines of evidence showing that the observed interactions are independent of cell density:

Many of these direct cellular surface interactions were observed for several hours without any interruptions, thus it is unlikely that these were random contacts simply enforced by the high density of microglial and AKT1 positive cells. To address this further, we provided an additional data set in the previous paper. Here, we injected lower amounts of the lexOP:*AKT1*-lexOP:tagRFP construct into oocytes of double transgenic mpeg1:EGFP/NBT:ΔLexPR fish and screened for fish with lower numbers and isolated AKT1 positive cells. In these larvae microglia were frequently seen to interact with AKT1 positive cells (previous paper (Chia et al., 2018): Figure 3C, D; Video 3 and 4, n = 6 samples analyzed). These interactions lasted for several hours and were not seen with control cells (not shown). Thus, we conclude that microglia are pro-actively interacting with AKT1 positive cells.

We would like to highlight that all quantifications of interactions in this manuscript were performed at 5 dpf. At this developmental age larval zebrafish microglia appear to be sessile and random movements are significantly reduced compared to earlier developmental time points (3 dpf and 4 dpf) (Svahn et al., 2012, Figure 5B). Thus, it is unlikely that microglia randomly contact AKT1 cells because of their movements.

Furthermore, for the quantifications provided in this manuscript we have normalised for the increased number of microglia in AKT1 samples compared to control samples. The quantifications show the percentage of microglia which displayed interactions with RFP control cells or AKT1 positive cells (number of interacting microglia/number of total microglia x 100). We have added an additional line in the Results and extended our explanation in the Materials and methods now to highlight this point.

The number of AKT1 cells or RFP control cells varies between samples due to the transient nature of the expression system. We have quantified the number of AKT1 cells and RFP cells in the analysed 5dpf samples and do not detect a significant difference between control and AKT1 samples. The average number of AKT1 cells in these samples (at 5 dpf) was 383 while the average number of RFP cells was 519. Increased proliferation rates in AKT1+ cells are detected from 7 dpf and lead to increased numbers of AKT1+ cells only at later stages. Thus, as numbers of RFP cells and AKT1 are comparable at 5 dpf, this does not impact on possible interactions.

I wonder about the specificity of this interaction (although the authors do show videos of the microglia interacting with/directly touching the neural cells) – could they show another cell type in the brain that isn't contacted in the same way as the neural cells?

The responses observed appear to be highly specific for cells undergoing malignant transformation. While control RFP neural cells are only sporadically contacted by microglia we see a significant increase in the number of interactions with AKT1 positive neural cells. This is also shown in the Videos 1-4 of the previous manuscript.

We have now overexpressed the HRASV12 oncogene in neural cells, which stimulates similar microglial responses (Figure 1—figure supplement 1). Furthermore, we have expressed AKT1 and HRASV12 under the zic4 enhancer in proliferating domains of the developing CNS (Distel et al., 2009, Mayrhofer et al., 2017). Here we observe an attraction of microglia to HRASV12 and AKT1 positive zic4 cells and direct cellular interactions as well (Figure 1—figure supplement 1). We conclude that these interactions are induced during the process of oncogenic cell transformation, independent of the cell type in the CNS. Future studies will reveal if differences between oncogenes and cell types exist.

3) The authors claim that AKT1+ cells display more frequent Ca^2+^ firing and show nice examples (2E, 2F) but a quantification should be provided.

We have not provided quantifications here as the observed Ca^2+^ firing (∆F/F_0_ increase > 0.05) in AKT1 cells has not been observed in any of the analysed control RFP cells. In contrast, 31% of the observed AKT1 cells show these Ca^2+^ firing events. The analysed control RFP cells showed only minor changes in their Ca^2+^ levels (∆F/F_0_ increase < 0.03). We have included this information in the text now. We speculate that these significant increases in Ca^2+^ are caused by the malignant transformation due to AKT1 overexpression and cannot be reached under physiological conditions.

4) Are tumor cells with the highest Ca^2+^ levels more frequently in contact with microglia? This would certainly reinforce the author's claims. This may not be trivial as the Ca^2+^ reporter use the same fluorescence channel as the macrophage reporter, but the patterns are very different so it should be doable.

This is indeed a very interesting question but not trivial to address as the Ca^2+^ reporter and the macrophages/microglia are visualised in the same channel. Furthermore, signal levels for the Ca^2+^ reporter are significantly lower compared to signal levels from macrophages/microglia. Thus, imaging settings need to be carefully adjusted to visualise Ca^2+^ levels but not to massively overexpose macrophages/microglia. We have tested this now and managed to visualise macrophages/microglia and high-level Ca^2+^ cells at the same time. We provide a new set of images in the revised version of the manuscript (Figure 3). These images show examples of microglia directly responding to cells upon an increase in Ca^2+^ levels.

5) It is satisfying to draw a straight line from AKT1 activation to Ca^2+^ to ATP release to p2ry12 receptor. The logic seemed to be that increased calcium leads to ATP release which would link to microglial interactions. But, why would increased AKT1 activity lead to increased intracellular Ca^2+^? Is this a direct effect, or are there expected to be many steps between AKT1 and Ca increase? Does AKT1 have other important effects? If Ca^2+^ is the primary mediator of the effect, can they induce increased Ca^2+^ in neurons via another mechanism (e.g. drug treatment or genetic) and see a similar effect?

Indeed, based on our data AKT1 activation results in increased Ca^2+^ levels, which leads to ATP release and P2ry12 activation. AKT1 signalling is activated in many cancers and governs tumour initiation and progression. AKT1 signalling is involved in a variety of cellular functions such as proliferation, metabolism, cell survival and migration. However, how AKT1 impacts on Ca^2+^ levels (directly or indirectly) has not been addressed so far. We speculate that the increase in Ca^2+^ levels is not specific to the oncogene (AKT1 in this case) but is part of the process of oncogenic transformation. We have tested HRASV12 now in addition and see similar increases in Ca^2+^ levels in HRASV12 positive cells. Future studies will have to reveal how oncogenic transformation leads to an increase in Ca^2+^ levels. As we highlight in the discussion of the manuscript, astrocytoma cells have been shown to form networks and to communicate via Ca^2+^ transients (Osswald et al., 2015). Ongoing work in the laboratory is currently addressing if the Ca^2+^ transients observed here resemble the first signs of network formation.

Ca^2+^ is clearly the primary mediator of the observed interactions. We and others have shown in previous publications that an increase in Ca^2+^ levels in neural cells is sufficient to stimulate microglia interactions. In Sieger et al., 2012, we have injected caged IP3 into single hemispheres of larval brains. Local flash-uncaging of IP3 resulted in increased Ca^2+^ levels in single neurons. This resulted in responses of surrounding microglia sending their processes towards the uncaged spot (Sieger et al., 2012, Figure 3G, H and Figure 5 B-D). Furthermore, we could show that these responses to local Ca^2+^ increase were dependent on ATP and P2ry12 signalling (Sieger et al., 2012, Figure 5E-J). These results were confirmed by others using different methods. Li et al., 2012, used local uncaging of glutamate in larval zebrafish brains and reported similar responses of microglia. Eyo et al., 2014, used a slice culture model and showed that NMDA mediated Ca^2+^ increase in neurons directly attracts microglial processes. In conclusion, there is published evidence using a variety of models and tools showing that the pure increase in Ca^2+^ levels in neural cells is sufficient to mediate microglial processes.

6) Could loss of p2ry12 in microglia lead to other deficiencies/abnormalities in the microglia? They have similar numbers of microglia per the author's data, but do they move around as much? Do they extend the same number of processes? One might predict that increased p2ry12 expression on microglia might lead to increased interactions with neurons (even in the absence of AKT1+ transgene expression with "basal" ATP release) and then even potentially increased neural cell numbers, if this is a primary mechanism by which microglia induce neural cell number increase.

This is a very interesting question which has been addressed by us and others in recent years. In Sieger et al., 2012, we used morpholino oligonucleotides and pharmacological tools to inhibit p2ry12 function. We could show that injection of p2ry12 specific morpholinos (as well as pharmacological inhibition) completely abolished microglial responses to injury mediated Ca^2+ –^ ATP signalling. In line with these studies we showed that other microglial functions were not altered upon p2ry12 knockdown. Microglia motility, process length and process motility as well as their capacity to respond to bacteria and phagocytose bacteria were similar to controls (Sieger et al., 2012, Figure S1 and S2).

Haynes et al., 2006 (Nature Neuroscience) analysed p2ry12 deficient mice and showed that these mice have a normal prevalence, distribution and morphology of microglia. Furthermore, they showed that processes from p2ry12 deficient microglia occupy the same area as processes from wt microglia and that baseline process motility of p2ry12 deficient microglia is not altered (Haynes et al., 2016, Supplementary Figure 1). Thus, according to our own published data as well as published data from rodent models, P2ry12 signalling is specifically needed for microglial responses towards extracellular ATP.

Interestingly, under physiological conditions, ATP-p2ry12 mediated contacts between microglia and neural cells do not impact on the proliferative capacity of neural cells. In Figure 5B of the current manuscript we show that proliferation rates of neural cells in p2ry12 deficient larvae are similar when compared to control larvae. Li et al., 2012, identified the role for microglial process contacts under physiological conditions and showed that highly active neurons stimulate microglial contacts via ATP release and that these contacts lead to a downregulation of neural activity. Thus, the identified role for microglial processes to induce proliferation seems to be specific for neural cells undergoing malignant transformation.